# Qualitative study exploring factors affecting the implementation of a vocational rehabilitation intervention in the UK major trauma pathway

Jade Kettlewell ,[1] Kate Radford ,[2] Denise Kendrick,[1] Priya Patel,[3] Kay Bridger,[4] Blerina Kellezi ,[1,4] Roshan Das Nair,[3] Trevor Jones,[1] Stephen Timmons,[5] On behalf of the ROWTATE team

For numbered affiliations see end of article.

**Correspondence to**
Dr Jade Kettlewell;
Jade.Kettlewell2@nottingham.ac.uk

## ABSTRACT

**Objectives** This study aimed to: (1) understand the context for delivering a trauma vocational rehabilitation (VR) intervention; (2) identify potential barriers and enablers to the implementation of a VR intervention post-trauma.

**Design** Qualitative study. Data were collected in person or via phone using different methods: 38 semistructured interviews, 11 informal 'walk-through care pathways' interviews, 5 focus groups (n=25), 5 codesign workshops (n=43). Data were thematically analysed using the framework approach, informed by the Consolidated Framework for Implementation Research.

**Setting** Stakeholders recruited across five UK major trauma networks.

**Participants** A variety of stakeholders were recruited (n=117) including trauma survivors, rehabilitation physicians, therapists, psychologists, trauma coordinators and general practitioners. We recruited 32 service users (trauma survivors or carers) and 85 service providers.

**Results** There were several issues associated with implementing a trauma VR intervention including: culture within healthcare/employing organisations; extent to which healthcare systems were networked with other organisations; poor transition between different organisations; failure to recognise VR as a priority; external policies and funding. Some barriers were typical implementation issues (eg, funding, policies, openness to change). This study further highlighted the challenges associated with implementing a complex intervention like VR (eg, inadequate networking/communication, poor service provision, perceived VR priority). Our intervention was developed to overcome these barriers through adapting a therapist training package, and by providing early contact with patient/employer, a psychological component alongside occupational therapy, case coordination/central point of contact, and support crossing sector boundaries (eg, between health/employment/welfare).

**Conclusions** Findings informed the implementation of our VR intervention within the complex trauma pathway. Although we understand how to embed it within this context, the success of its implementation needs to be measured as part of a process evaluation in a future trial.

## Strengths and limitations of this study

► This study provides new evidence to support the implementation of a vocational rehabilitation intervention in the trauma population, presenting the views of 117 key stakeholders.

► Recruitment of a diverse sample of service users and providers across different National Health Service sites and areas of the UK, providing a broad perspective on the factors affecting implementation.

► Use of a range of methodologies enabling in-depth discussion with key stakeholders.

► Unable to recruit employers to interviews or focus groups, thus the employer perspective is not presented.

## INTRODUCTION

Traumatic injuries are particularly problematic in working age adults, with road traffic injuries being one of the highest causes of death among 15–49 years old.[1] Survival rates for traumatic injuries have improved,[2] increasing the number of working age adults living with the long-term effects of injuries. These include physical and psychological problems, mental health conditions or hidden disabilities (eg, urological/cognitive problems), which affect ability to return-to-work (RTW).[3–10] Injured patients may benefit from RTW support addressing physical and psychological needs. However, provision of RTW support, known as vocational rehabilitation (VR), is inconsistent across the UK[11–13] and known issues (eg, poor communication between acute/community care, long waiting lists) in the major trauma pathway make access to VR and psychological support challenging.[11]

VR is a complex intervention,[14] and requires the coordination of multiple systems, across health, social care and employing

organisations. Complex interventions are difficult to evaluate in clinical trials because they may not be delivered consistently and are context-specific. The UK Medical Research Council advocates developing complex interventions systematically, by involving stakeholders, using evidence-based theory, and testing them in a phased approach[14] to ensure successful delivery in clinical practice.[15] Understanding intervention context is essential,[16] and researchers are encouraged to consider how an intervention is expected to work (ie, the internal intervention logic). Overall systems, including which parts of the system could influence the intervention, and how the intervention could lead to wider system change,[16] should be considered.

Current research evaluating the delivery of VR interventions tends to focus on one health condition or one location,[17–19] but this limited evidence does not account for the complexity of trauma or the National Health Service (NHS) trauma rehabilitation pathway, thus limiting the identification of barriers and facilitators to VR delivery.

As part of a research programme to develop and evaluate a VR intervention for major trauma patients,[20] we set out to explore the potential implementation issues associated with the delivery of this intervention across five UK major trauma centres. It was important that we understood how the intervention could be implemented across different injury types and NHS Trusts to ensure our intervention design was robust ahead of a definitive trial.

ROWTATE (www.ROWTATE.org.uk) is an individually-tailored job retention intervention, delivered by an occupational therapist (OT; acting in case-coordinator role) and clinical psychologist, to people with at least 'moderate trauma' (defined as an Injury Severity Score of 9 or greater.[21] It commences within 12 weeks post-injury and lasts up to 1 year. It involves assessing the impact of the injury on the person and job role, rehabilitation to prepare the patient for work, plan/monitor a phased RTW by liaising with employers and the healthcare team, educating patients/employers about injury impact, and early identification, monitoring and support for psychological problems.

The aims of this study were to: (1) understand the context for delivering the ROWTATE VR intervention and (2) identify potential barriers and enablers to the implementation of VR following major trauma in a diverse trauma injury population. The findings of this study informed the development of the ROWTATE intervention and considerations for implementation ahead of a future trial.

## METHODS

This multiplemethods qualitative study was part of a larger programme of work funded by the National Institute for Health Research (RP-PG-0617-20001).

## Participants

We recruited key stakeholders (individuals that provide, deliver, or receive trauma/VR, or work/care for trauma survivors) including trauma service users, carers, NHS service providers, private rehabilitation providers, third sector services and the insurance industry. NHS service providers worked across different settings: acute, community and primary care, and included general practitioners, trauma rehabilitation specialists and psychologists.

Stakeholders were recruited in five UK major trauma centres in 2019–2020 using purposive sampling. We chose this method to ensure we recruited key stakeholders within each major trauma centre that had a clear understanding of the rehabilitation pathways (based on a priori knowledge of clinical expertise in each site), and to ensure different injury types, pre-injury occupations (including self-employed), socioeconomic backgrounds, geographical locations, and different professionals (including employers) were represented in our sample. Potential participants were identified via known contacts of the authors at the different major trauma centres, who were asked to share the email invitation with relevant colleagues or trauma patients (ie, snowball sampling). Service users were identified via trauma patient and public involvement (PPI) groups at the University of Nottingham and via a database of previous patient participants (from other studies) that had agreed to be contacted about future research. Trauma participants and carers were offered a £20 gift voucher for their time, plus travel expenses where necessary.

## Data collection

Data were collected using semistructured interviews, focus groups, codesign workshops and informal 'walk-through care pathways' interviews (ie, walking interviews conducted in a major trauma centre or repatriating site with an appropriate individual to better understand the context for delivery in that site). We aimed to recruit up to 195 participants (focus groups n≤40; interviews n≤20; walk-through care pathways n≤60, up to 12 per site; workshops n≤75, up to 15 per site), however stopped recruitment when we reached theoretical sufficiency.[22] Due to the COVID-19 pandemic, we were unable to conduct as many walk-through care pathways interviews as proposed, thus conducted additional semi-structured interviews.

All participants were informed about the aim of the study and their right to withdraw. Written informed consent was obtained from all interviews and focus groups. For codesign workshops and walk-through care pathways, consent was assumed if participants did not opt out. Data collection took place in participants' places of work (including patient participants if necessary) or university premises. Topic guides for each method of data collection were developed following analysis of previous research with VR and trauma patients conducted by the authors and in discussion with ten PPI members. Table 1 provides a summary and examples of each topic guide.

**Table 1** Summary of topic guides

| Topic guide | Focus of activity | Example of questions |
|---|---|---|
| 1. Interview and focus group topic guide for service providers | ▶ Discussing experiences and opinions of current services<br>▶ Identifying any service gaps that exist.<br>▶ Describing proposed return to work intervention/programme called ROWTATE and ask for feedback.<br>▶ Identifying any potential barriers to delivery within the NHS. | ▶ Does your organisation currently provide return to work services/support for people after trauma?<br>▶ Thinking about the needs of people after trauma, where do you think there are service gaps?<br>▶ Is there an unmet need for vocational support after injury?<br>▶ Which trauma related problem(s) (eg, physical health, mental health, other) should our return to work programme focus on?<br>Thinking more specifically about the proposed ROWTATE programme…<br>▶ What things need to be in place to allow the programme to begin (resources)<br>▶ Who should provide the programme and what training will they require?<br>▶ Does the implementing organisation have the capacity to implement this programme?<br>▶ Will the clients face any barriers to receiving the programmes?<br>▶ What outcomes will be achieved by the intervention/programme?<br>▶ What environmental factors might work to support or act against implementation of the programme? |
| 2. Interview and focus group topic guides for service users (trauma and carer participants) | ▶ Discussing the impact of injury<br>▶ Experiences and opinions of current services<br>▶ Discussing gaps in services that were (or were not) available post-injury.<br>▶ Discussing return to work services, their purpose and why support isn't always provided/barriers to delivery.<br>▶ Presenting/describing the proposed return to work programme called ROWTATE and asking for feedback about content and potential barriers to delivery. | ▶ In your experience what services are available to support people who have major injuries?<br>▶ What are the issues people who have major injuries face in returning to and remaining in work?<br>▶ Thinking about people of working age who have major injuries, is there a need for services that support people in a return to work?<br>  If so, what should this service look like?<br>Thinking more specifically about the proposed ROWTATE programme…<br>▶ How does this programme fit with your ideas of what is needed? Will it address the problem?<br>▶ Can you think of anything that might prevent this programme from working?<br>▶ Can you think of any barriers to engaging in the ROWTATE programme?<br>▶ Do you think there may be any negative consequences?<br>  → For the injured person?<br>  → For the employer?<br>  → For the health service? |

NHS, National Health Service.

Focus groups were conducted by JK and KB with 15 trauma survivors, one carer and nine service providers. We (JK, KB, PP) conducted 38 semistructured interviews with ten trauma survivors, 1 carer and 27 service providers and 11 'walk-through care pathways' interviews with service providers from three Major Trauma Centres. We (JK and ST) undertook five codesign workshops with five trauma survivors and 38 service providers at five major trauma centres. See table 2 for recruitment summary and topics covered by each activity. A summary of participant characteristics is shown in table 3. A summary of researcher characteristics can be found in table 4. Researchers did not know participants prior to conducting research activities.

## Data analysis

All semistructured interviews and focus groups were audiorecorded and transcribed. For informal interviews and workshops, notes were taken. All data were thematically analysed using the framework approach[23] and coded using NVivo (by JK and PP), informed by the Consolidated Framework for Implementation Research (CFIR).[24] The main domains of CFIR were used for coding, and key themes were agreed by discussion with other authors (KR, ST) and further discussed with a PPI member (TJ). Barriers and enablers to implementation were identified across the interviews, focus groups and codesign workshops, then mapped, where possible, onto CFIR constructs. The Consolidated Criteria for Reporting Qualitative Research checklist has been used to ensure comprehensive reporting of our study (see online supplemental material 1).

## Patient and public involvement

PPI representatives were involved throughout this study. A patient representative (TJ) contributed to

 

**Table 2** Summary of participant recruitment by activity

| Activity | Purpose/topics covered | Average length of activity | Participant type | n | Total per activity (n=117) |
|---|---|---|---|---|---|
| Focus groups (n=5) | Psychosocial context of trauma survivors, essential resources needed for, and barriers to the implementation of a VR intervention. | 90 min | Trauma survivor | 15 | 25 |
| | | | Service provider | 9 | |
| | | | Carer | 1 | |
| Semi-structured interviews | Experiences of receiving or providing rehabilitation, understanding usual care and local unmet need, specific service gaps and lack of support, contextual factors affecting the implementation of a VR intervention. | 60 min | Trauma survivor | 10 | 38 |
| | | | Service provider | 27 | |
| | | | Carer | 1 | |
| Walk through care pathways | | 20 min | Service provider | 11 | 11 |
| Workshops (n=5) | Discussions about the VR intervention logic model, the local context for delivery and other factors that may affect its implementation. | 120 min | Trauma survivor | 5 | 43 |
| | | | Service provider | 38 | |

VR, vocational rehabilitation.

the development of this study proposal (and overall programme grant). A larger group of patient representatives (who form the ROWTATE PPI group) were involved in developing patient facing documents, including patient information sheets. Topic guides were developed alongside 11 trauma survivors (diverse group including 7 males, 4 females; self-reported injury: spinal cord injury (n=2,) traumatic brain injury (n=8), multiple fractures (n=5), polytrauma (n=1); pre-injury occupation: professional (n=6), managerial (n=2), student (n=1), military (n=1). PPI representative (TJ) was involved in data analysis.

**Table 3** Characteristics of participants

| Participant type | Professional role or injury type | Total (n=117) | Other demographic information |
|---|---|---|---|
| Service user (n=32) | Amputation | 1 | Gender: Male (n=15, 47%); Female (n=17, 53%) Pre-injury occupation: Self-employed (n=5, 16%); Employed (n=25, 78%); Student (n=2, 6%) Ethnicity: White British (n=30, 94%); Asian (n=1, 3%); Black British (n=1, 3%) |
| | Brain injury and poly-trauma | 13 | |
| | Carer | 2 | |
| | Orthopaedic injury | 13 | |
| | Spinal injury | 3 | |
| Service provider (n=85) | Case manager | 3 | Gender: Male (n=29,34%); Female (n=56, 66%) |
| | Clinical psychologist | 10 | |
| | Disability employment advisor | 3 | |
| | Doctor/physician | 16 | |
| | General practitioner | 4 | |
| | Occupational physician | 1 | |
| | Occupational psychologist | 1 | |
| | Occupational therapist | 27 | |
| | Physiotherapist | 5 | |
| | Psychiatrist | 1 | |
| | Solicitor | 2 | |
| | Speech and language therapist | 1 | |
| | Trauma charity coordinator | 2 | |
| | Trauma practitioner | 5 | |
| | Trauma rehabilitation coordinator | 1 | |
| | Trauma psychologist/psychotherapist | 3 | |

**Table 4**  Summary of researcher characteristics

| Characteristic | Researcher 1 (JK) | Researcher 2 (PP) | Researcher 3 (KB) | Researcher 4 (ST) |
|---|---|---|---|---|
| Gender | Female | Female | Female | Male |
| Education | MSc, PhD | MSc | MSc | MSc, PhD |
| Ethnicity | White British | Asian British | White British | White British |
| Research role/title | Research fellow | Research Assistant | Research Assistant | Professor |
| Experience | Traumatic injury research, rehabilitation psychology and implementation | Developmental and neuropsychology | Trauma psychology | Health services management, implementation |
| Research activity | Interviews Focus groups Codesign workshops Walk-through care pathways | Interviews | Interviews Focus groups | Codesign workshops |

## RESULTS

A visual summary in figure 1, provides an overview of factors that may affect the implementation of ROWTATE. Key barriers and facilitators are presented in the next section. Table 5 summarises findings specific to codesign workshops.

### Outer setting

#### Cosmopolitanism

This theme refers to the degree to which an organisation is networked with other external organisations.[24] We also considered transition between different organisations. Barriers identified by stakeholders were: inconsistent service provision across the major trauma pathway and poor communication between organisations when patients left the hospital. Many stakeholders spoke about the importance of supported transition across the healthcare pathway; however, some patients were often left with little or no support, especially on discharge from the acute setting:

> Once I left the hospital to go home, I had no support. I used to sit in my chair and just let my leg hopefully slowly mend which is what it did. Everything I was taught worked, but I had no support (Trauma survivor, orthopaedic injury)

Trauma survivors often left hospital with long-term problems requiring rehabilitation. Therapists referring patients onto other services were often left not knowing

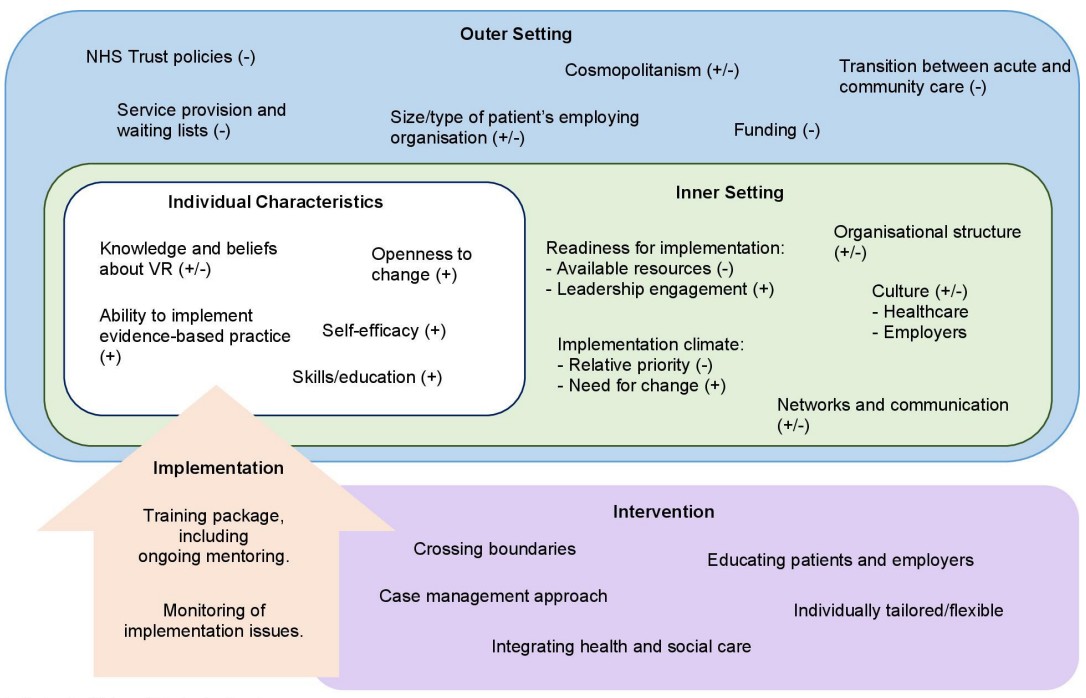

(+) indicates facilitator   (-) indicates barrier

**Figure 1**  Summary of barriers and facilitators to the implementation of a vocational rehabilitation intervention, mapped onto CFIR. CFIR, Consolidated Framework for Implementation Research; NHS, National Health Service; VR, vocational rehabilitation.

**Table 5** Summary of findings from codesign workshops mapped onto CFIR headings

| | CFIR constructs | Definition of construct | Key points made during codesign workshops |
|---|---|---|---|
| Outer setting | Patient Needs and Resources | The extent to which patient needs, as well as barriers and facilitators to meet those needs, are accurately known and prioritised by the organisation. | Some community rehabilitation teams were already providing VR and/or psychological support, however waiting lists are long meaning patients' needs are not always addressed in a timely manner. Additional resources and therapists would increase capacity, thus supportive of out intervention. |
| | Cosmopolitanism | The degree to which an organisation is networked with other external organisations. | Major trauma centres had good links with repatriating hospitals and community teams, however stakeholders highlighted the gap in communication between acute and community care. This was highlighted as a potential barrier to implementation. |
| | Peer Pressure | Do organisations feel peer pressure to adopt the intervention? | All participants were open to implementing the intervention in their NHS sites, however as services and processes are influenced by funding/commissioning, stakeholders felt this might be a barrier to long-term implementation. |
| | External Policy and Incentives | External strategies to spread interventions including policy and regulations, external mandates, recommendations and guidelines. | Stakeholders stated that policies may be a barrier to long-term implementation, but not a barrier in terms of study delivery. |
| Inner setting | Structural Characteristics | How the organisation works. The social architecture, age, maturity, and size of an organisation. | Stakeholders were open to change and felt our intervention would work well within their organisation if barriers addressed. |
| | Networks and Communications | The nature and quality of formal and informal communications within an organisation. | Communication between healthcare professionals within the organisation and multi-disciplinary working would facilitate intervention delivery. |
| | Culture | Norms, values, and basic assumptions of a given organisation. | Rehabilitation stakeholders appeared open to the implementation of a vocational intervention and felt it was an important intervention. |
| | Implementation Climate | Absorptive capacity for change, shared receptivity of involved individuals to an intervention and the extent to which use of that intervention will be rewarded, supported and expected within their organisation. | Stakeholders agreed intervention was important for people after trauma and supported its implementation, with the hope that their organisation would encourage its delivery long-term. |
| | Readiness for Implementation | Tangible and immediate indicators of organisational commitment to its decision to implement an intervention. | NHS sites ready to implement the intervention for the trial. |
| Characteristics of individuals | Knowledge and Beliefs About the Intervention | How much do stakeholders know about the intervention and what do they think about it. | Stakeholders agreed that the components of the intervention were appropriate and would be feasible to deliver if service specific barriers addressed. |
| | Self-efficacy | Individual belief in their own capabilities to execute courses of action to achieve implementation goals. | Stakeholders believe intervention is important and wanted to support its implementation in their NHS sites. |
| | Individual Stage of Change | Characterisation of the phase an individual is in, as he or she progresses toward skilled, enthusiastic and sustained use of the intervention. | Stakeholders enthusiastic about the intervention and keen to be involved. |
| | Other Personal Attributes | A broad construct to include other personal traits such as tolerance of ambiguity, intellectual ability, motivation, values, competence, capacity and learning style. | Stakeholders seemed motivated to implement the intervention in their different NHS sites. |

CFIR, Consolidated Framework for Implementation Research; NHS, National Health Service; VR, vocational rehabilitation.

whether that individual had received support, thus making it difficult to ensure continuity of care. There appears to be a lack of communication and networking across services:

> You feel responsible, and you want it to be right but, you do just sometimes have to say, "Right, I've passed on to that service." And you want that service to be perfect, but they are no longer your patient and it's not your right to know what happens to them. (Trauma physiotherapist)

Developing stronger links between organisations facilitates implementation. One stakeholder highlighted

the need for healthcare providers to communicate with a patient's employer to ensure awareness of their employee's injury and limitations. Crossing boundaries between organisations may be necessary to ensure patients are supported:

> There should be good communication between the healthcare providers…if that means a letter from the consultant (physician) to this person's employer saying, this is this person's injury… this is the extent of the difficulties this person is going to endure. (Trauma psychotherapist)

Stakeholders also suggested the need for early contact with employers to explain the impact of their employee's injury and recovery trajectory, facilitating job retention:

> Therapists feeling confident to talk about work and quite early on. Even if you're not necessarily doing something about it, but asking the question, finding out what their job involves. … having that contact to say, this is kind of the estimated length of time and just keeping them (employer) in the loop so that they don't lose their job as a result. (Vocational OT)

### Structural characteristics

This theme refers to how the employing organisation works; its social architecture, age, maturity and size.[24] The ability for an individual to successfully RTW may be related to the type and size of their employing organisation. Stakeholders stated it is often difficult to get patients back to their pre-injury role (eg, physical jobs), or that the company is too small to make reasonable adjustments for the injured person:

> Some of our companies will support people over months, but other companies who are much smaller or have very specific, very physical jobs, it's very hard for them to accommodate a large number of adjustments. (Occupational physician)

### External policy and funding

This theme refers to external strategies to spread interventions, which included policy, regulations, recommendations and guidelines.[24 25] Service providers discussed issues when having no central point of contact within the trauma pathway and how constantly changing services posed a barrier for effective rehabilitation. Service providers were frustrated with decommissioning of services, low prioritisation of rehabilitation services and lack of clarity on what funding is available, particularly for trauma survivors with mental health problems:

> It's [rehabilitation] the first thing to get axed when budget cuts come in. (Trauma rehabilitation consultant)

> Commissioning's stagnant at the moment… There's no money for mental health. (Trauma psychotherapist)

Service providers and trauma survivors frequently mentioned the lack of rehabilitation services across areas of the UK, mainly in areas located further from major trauma centres:

> It is quite postcode lottery-type thing. Some places have generic outpatient departments, so there is no specific neuro or amputee or spinal expertise and that's the same for community rehab. (Case manager)

Therapy teams were overstretched with long waiting lists, so introducing new interventions was a concern, particularly for therapy managers. Other issues related to how external policies allowed for cross-boundary working, (eg, therapists accessing clinical notes from other NHS Trusts, or liaison with the Department for Work and Pensions or employers):

> If your substantive role is on the acute sector then crossing boundaries to the community and out to the employers…or vice versa. (Vocational OT)

### Inner setting
#### Culture
This theme refers to the norms, values, and basic assumptions of an organisation,[26] either healthcare or employing organisations. Healthcare professionals and employers may consider that the long-term effects of some injuries (eg, brain injury) are too severe for patients to RTW, preventing VR access:

> I suspect that some people [healthcare providers] think there's nothing you can do about a brain injury, so what's the point in referring [for VR]. (Trauma clinical psychologist)

Participants argued that there is a culture within acute care units to discharge patients when they are medically fit for discharge, which may lead to individuals being discharged without the necessary vocational and psychological support:

> An acute hospital will never be geared up for those [vocational and psychological] needs. Because their priority is, getting people through the medical system, assess, treat, [and] discharge. (Consultant neuropsychologist)

Healthcare professionals within the acute setting were concerned about identifying potential problems that an individual was experiencing because of lack of services to which to refer:

> There's just not really the [vocational or psychological] services out there to then signpost people on to. So, you almost feel like you're opening a Pandora's box where you can't actually then put those pieces back in. (Trauma OT)

Consequently, trauma patients may be sent home with a rehabilitation plan that did not reflect their needs. Participants also discussed long-standing barriers of

hierarchy and a lack of understanding about allied health professional roles. One therapist stated that healthcare providers other than doctors were often ignored by employers, making an OT-led intervention difficult to implement:

> Sometimes they [employers] will not take the views of OTs, physios, speech and languages, as seriously as they should, and sometimes they want to see doctor after their name. (Consultant neuropsychologist)

Stakeholders also recognised potential facilitators, such as the need for changing the 'norm' within the NHS and ensuring therapists and managers were committed to the intervention. One therapist highlighted the need to translate research into the real-life context of the healthcare system:

> It's not just the development or the intervention itself, it's actually the integration of it into the culture so that it changes the practice of what people think of as the norm… and getting a lot of people to buy into it. (Senior trauma physiotherapist)

### Implementation climate

This refers to the capacity for change and the extent to which use of that intervention would be rewarded, supported and expected within their organisation.[27 28] Stakeholders could see the need for change and were supportive of the intervention, as current services provided limited vocational support. However, organisations both inside and outside the hospital did not see VR as a priority relative to other services.

Major trauma centres lacked awareness and knowledge that RTW or education might be an important part of long-term recovery. Healthcare providers in the acute setting were not routinely asking about RTW, so patients were discharged without being offered the early support they may benefit from:

> Some people have gone out [of hospital] before four weeks, and they would maybe be people who would benefit from some vocational input…they've got cages on their legs, they've got to have skin grafts, so their return-to-work could be six to twelve months, but they need to have that question [what is your job?] asked. (Vocational OT)

Sometimes referral to specialist VR services did not happen as healthcare professionals did not feel RTW was going to be problematic. Again, this related to a lack of knowledge in the acute setting about the impact of injury on ability to RTW.

> I think it was very surprising how few people were referred to this [VR] clinic…it seems to be me that there was a kind of, yes return-to-work is important, but a complete lack of understanding of, 'there's going to be a problem'. (Consultant neuropsychologist)

Employers and patients also lacked knowledge and awareness about the importance of vocational support:

> There may be massive difficulties or concerns around going back to work, those don't come to light until they've become an outpatient…they haven't had contact with their employers and their job may have already come to an end. Whereas if you'd had an intervention earlier that could have been saved. (Vocational OT)

There is a known stigma about mental health[29–32] and patients are sometimes concerned about disclosing psychological problems to employers, which can prevent them receiving support. Therapists highlighted concerns over patients declaring non-visible conditions to employers (eg, anxiety, depression, pain). However, such non-disclosure may prevent individuals receiving the RTW support they needed:

> There's a stigma about talking about mental health. Is there any chance that by encouraging somebody to discuss their mental health you are actually harming their future employment? (Trauma psychologist)

### Readiness for implementation

This theme is defined as tangible and immediate indicators of organisational commitment to its decision to implement an intervention.[24] Service providers discussed their difficulties in working flexibly and outside their normal area of expertise, including working across areas of healthcare they were less familiar with, or having to liaise with employers:

> There are some people who just cannot escape their programming from their training… But in this kind of working the OT needs to do a bit of psychology and the psychologist needs to do a bit of OT. And if the psychologist's too precious to do OT then that's not going to work very well. (Clinical psychologist)

### Characteristics of individuals
#### Self-efficacy

This theme refers to an individual's belief in their own capabilities to execute courses of action to achieve implementation goals.[33] For the delivery of a VR intervention to be successful, the patients needed to believe in their ability to RTW, and the employer needed to believe that they could support them. Service providers and patients talked about the importance of changing employer attitudes towards traumatic injury:

> Part of the difficulty is trying to work out how you can change work and getting the employers to think differently about why they should support somebody going back to work, particularly in high demand and highly technical jobs or very physical jobs. (Occupational physician)

An employer's desire to act on the advice of a therapist or make reasonable adjustments to support an RTW could become a barrier to VR intervention implementation:

There wasn't really an option to sit down, so I asked why I couldn't have a chair, they [employer] said, well you can't obstruct the walkway…they weren't that understanding really. (Trauma survivor, brain injury)

Feedback from stakeholders suggested that most employers were willing to accept advice and improve understanding about injury to better support their employee. This facilitated VR delivery, as supportive employers were invested in the RTW process:

The good employers will make the reasonable adjustments, they will understand the situations, they will go and ask the necessary questions and do the best to support them. (Disability employment advisor)

Some patients felt they were 'damaged' and lost confidence in their abilities, which impacted motivation to RTW, or ability to push themselves:

I am not pushing myself forward [within work] as well because I think I am damaged goods. (Trauma survivor, spinal cord injury)

This posed an issue for the implementation of VR because patients needed to have the desire to RTW and to believe that they could. Therapists needed to work with patients to encourage them to work towards rehabilitation goals.

A key facilitator to delivering a successful VR intervention was ensuring patients understand the impact of their injury and how this would affect work, including the importance of not returning-to-work too soon:

I think I should have taken on some reduced duties or something first. That was my choice. They [employer] offered that and I said I'd be fine and then it turned out pretty bad for me. (Trauma survivor, orthopaedic injury)

### Knowledge and beliefs about VR

This theme refers to individuals' attitudes towards, and value placed, on the intervention.[24] When discussing the content of ROWTATE, stakeholders felt it was appropriate and could see the value in providing combined vocational and psychological support to trauma survivors.

One stakeholder highlighted the importance of providing a VR intervention to ensure patients do not RTW before they have fully recovered:

I've got many, many patients who ignore our [rehabilitation team] advice and go to work earlier and they go a step backwards…within a week will say they are back to work, and I am surprised how on earth they did that and what risk they are taking. (Rehabilitation physician)

There also appeared to be a gap in therapist knowledge about how to provide VR and/or a lack of confidence in asking questions about RTW. This posed an issue for delivering a VR intervention in practice, as therapists are not routinely trained to ask patients work-specific questions:

A lot of OTs aren't feeling confident about asking that [RTW] question, they find it quite scary. (OT)

Stakeholders mentioned that some employers have little understanding of injury, especially if they had never been faced with this situation, leading to anxieties about supporting an employee's RTW:

Sometimes employers are frightened about taking on something that they don't understand. (Case manager)

### DISCUSSION

Conducting implementation research in advance of the design of the intervention has meant that not only have we identified barriers to implementation, but we have also been able to actively address them. The key barriers were: cultural norms within healthcare and employing organisations, the extent to which healthcare systems were networked with other organisations, poor transition between different organisations, and failure to recognise VR as a priority, often as a result of policies and funding. Although some of these findings are more relevant within the UK context (eg, policies and funding), certain implementation barriers are applicable to global healthcare setting (eg, culture within the healthcare and employment sectors, cosmopolitanism and VR knowledge).

Though many of the issues that we have presented are classic implementation issues,[34] they represent substantial challenges for ROWTATE, which we have attempted to modify before it is implemented as part of a feasibility study.[20]

One substantial barrier was deep-seated cultural practices among a wide variety of stakeholders and beliefs about the need for early VR in the trauma population. Similar to Mannion and Davies,[35] we view culture as a complex and dynamic phenomenon which is not amenable to simplistic interventions. Our study highlighted the negative impact that certain cultural norms can have on the implementation of a complex intervention like VR. This issue of culture is consistently reported in the literature[34 36 37] as a barrier to implementation, with studies indicating that improving service providers' attitudes towards change,[38 39] encouraging flexibility in their way of thinking[40] and identifying champions[41 42] can facilitate intervention delivery. We have sought to address cultural issues within the intervention design by developing an in-depth training and mentoring package to explain and provide evidence for the importance of a VR early intervention in the trauma population[43–46] and increase therapist knowledge of VR. This training and mentoring package will be used for OTs and clinical

psychologists providing the intervention in the definitive trial.

Another key barrier was inadequacy of organisational networking, and the subsequent lack of communication and continuity of care across the trauma pathway. Often, trauma patients were discharged from the acute setting without any vocational or psychological support,[47] and when referred, therapists could not be confident that their patient would receive the support they needed. This lack of communication is well documented,[12 48–50] posing a barrier for VR implementation. Facilitating the link between health, employment, and others involved in the RTW process (eg, insurance industry, solicitors, case managers) may overcome some of these issues enabling a sustainable RTW. To address this issue, training for the OTs and psychologists who deliver our VR intervention positively encourages boundary-crossing. The ROWTATE training and intervention are predicated on a model where the OT acts as a case manager, facilitating communication between different stakeholders.

Findings from this study informed the programme theory and training package for the ROWTATE intervention feasibility study[20] and trial. Patients need a central point of contact when discharged from the major trauma centre or acute setting, to improve transition into the community, but also to communicate with key stakeholders involved in supporting RTW. A case coordinator, who also delivers the VR (eg, OT), is essential to supporting its implementation. Findings also suggested that early contact with an employer enables increased awareness of the impact of injuries on employees and their ability to work and to facilitate job retention. For a VR intervention to be implemented, the therapist should contact the patient within the acute setting (where possible), or soon after discharge. Crossing boundaries across different sectors is an essential part of a VR intervention, thus buy-in from healthcare professionals across the trauma pathway and from employers is necessary.

We drew on existing evidence to develop the employer engagement component of our intervention.[51 52] Previous research in acquired brain injury and spinal cord injury indicates that patients who understand the impact of their injuries are more able to discuss their limitations with employers.[17, 53] Understanding their employee's injury and limitations is essential in facilitating a successful RTW.[17 53 54] In general, employers feel they lack the necessary experience and knowledge to support an RTW for someone post-injury.[17 53] Further evidence suggests that employers require functional advice (eg, planning phased RTW, reasonable adjustments) as well as psychological support to understand and address the needs of their employee.[53 54] Providing a central point of contact to liaise with, educate and support the employer and patient, negotiating RTW plans, and providing advice and emotional support (via OT and clinical psychologist), are key components of our intervention.

Therapists need to be able to adapt to different circumstances and be flexible in intervention delivery for VR to be successful, as every trauma patient is different. To support implementation, a training package has been developed to train therapists to deliver the VR intervention, along with ongoing mentoring. Finally, findings identified a lack of psychological support for trauma patients. Thus, a VR intervention should include a psychological component, to ensure both physical and mental health issues are considered during RTW. For the definitive trial, where trauma services do not have access to psychological support, we will explore options for provision of psychological support from other NHS Trusts or from private practitioners.

We addressed as many issues as possible with the adaptations of the intervention (eg, improving communication by including a central point of contact/case manager), however some implementation issues are not easily overcome (eg, culture, organisational structure). Such barriers require long-term improvements, not only to change professional behaviours and cultural norms, but also to make small-step system changes to enhance coordination across the trauma pathway. While we adapted our intervention to overcome known barriers, we anticipate new implementation issues arising when the intervention is delivered as part of a trial (ie, real-world testing).

Our study had several strengths, providing new evidence to support the implementation of a VR intervention in the trauma population. We recruited a diverse sample of service users (including self-employed patient participants) and providers across different NHS sites and areas of the UK, thus presenting a broad perspective on the factors affecting implementation. Our findings triangulate information from different perspectives, using a range of methodologies, which enabled in-depth discussion and complex stories to be heard and understood. Our research team is comprised of practitioners (OTs, doctors) and multidisciplinary academics. However, we were unable to recruit any employers to participate in the interviews or focus groups, due to the COVID-19 pandemic and difficulty contacting employing organisations. This meant the views of these stakeholders were not explored in this context, and further work is required to understand their important perspectives on RTW post-trauma. Although we aimed to engage with as many different stakeholders as possible, we do feel some professions were underrepresented in our sample, including nurses, prosthetists, orthotists and trauma surgeons.

## CONCLUSION

Most implementation research continues to focus on one health condition and/or one context, limiting empirical understanding of the complex networks through which much contemporary healthcare is delivered. We addressed this gap by exploring issues across multiple trauma networks and conditions to enhance understanding of how the intervention could be implemented in different contexts and to ensure our intervention design and trial processes were appropriate. Identification of key barriers and facilitators to implementation has informed the development of the ROWATE intervention, which is ready to be tested in a trial across five UK major trauma pathways. Although we understand how best to embed the intervention within these complex systems, the success of its implementation will need

to be measured as part of a process evaluation. This will lead to a greater understanding of how the intervention might impact wider system change and factors affecting future widescale clinical implementation. Findings could illuminate similar barriers in other complex healthcare pathways outside of major trauma (eg, cultural norms, poor communication), and may inform intervention implementation within other conditions.

**Author affiliations**
[1]Centre for Academic Primary Care, University of Nottingham School of Medicine, Nottingham, UK
[2]Centre for Rehabilitation & Ageing Research, University of Nottingham School of Medicine, Nottingham, UK
[3]Institute of Mental Health, University of Nottingham School of Medicine, Nottingham, UK
[4]Department of Psychology, Nottingham Trent University, Nottingham, UK
[5]Centre for Health Innovation, Leadership & Learning, Nottingham University Business School, University of Nottingham, Nottingham, UK

**Acknowledgements** This paper has been written on behalf of the wider Return to Work after Trauma (ROWTATE) team, including all grant coapplicants and members of our Patient and Public Involvement (PPI) group. The authors would like to acknowledge their support in conducting this research. We would also like to thank all participants for taking part in this research.

**Collaborators** The following collaborators contributed to the acquisition of funding for the associated programme grant and/or are members of the programme management group:Professor Marilyn James, Professor Amanda Farrin, Ms Ivana Holloway, Dr Karen Hoffman, Mr Adam Brooks, Dr Matthew Smith, Ms Miriam Duffy, Professor Richard Morris, Dr Edward Carlton, Dr Judith Allanson, Dr Fahim Anwar.

**Contributors** JK: methodology, formal analysis, writing, data collection, review and editing; KR: conceptualisation, funding acquisition, project administration, methodology, formal analysis and review; DK: conceptualisation, funding acquisition, project administration, methodology, writing, review and editing; PP: methodology, formal analysis, data collection, review and editing; KB: methodology, formal analysis, data collection, review and editing; BK: funding acquisition, project administration, methodology, formal analysis, review and editing; RDN: funding acquisition, methodology, writing, review and editing; TJ: funding acquisition, methodology, formal analysis and review; ST: guarantor, funding acquisition, methodology, data collection, formal analysis, writing, review and editing; All authors have read and agreed to the published version of the manuscript.

**Funding** This paper presents independent research funded by the National Institute for Health Research (NIHR) under its Programme Grants for Applied Research Programme (Reference number RP-PG-0617-20001).

**Disclaimer** The views expressed are those of the authors and not necessarily those of the NIHR or the Department of Health and Social Care.

**Competing interests** None declared.

**Patient and public involvement** Patients and/or the public were involved in the design, or conduct, or reporting, or dissemination plans of this research. Refer to the Methods section for further details.

**Patient consent for publication** Not applicable.

**Ethics approval** Ethical approval was obtained from the University of Nottingham Faculty of Medicine and Health Sciences Research Ethics Committee (Ref: FMHS 150-1811) and Leicester South NHS Research Ethics Committee (Ref 19/EM/0114). Participants gave informed consent to participate in the study before taking part.

**Provenance and peer review** Not commissioned; externally peer reviewed.

**Data availability statement** Data are available on reasonable request. The data that participants have consented to share will become available to potential researchers at the end of this study. Requests detailing the research aims and use of the data should be sent to the research team via email: ROWTATE@nottingham. ac.uk.

**ORCID iDs**
Jade Kettlewell http://orcid.org/0000-0002-6713-4551
Kate Radford http://orcid.org/0000-0001-6246-3180
Blerina Kellezi http://orcid.org/0000-0003-4825-3624

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
