## [Reviewer comments · BMJ Open]

ARTICLE DETAILS

TITLE (PROVISIONAL)	A qualitative study exploring factors affecting the implementation of a vocational rehabilitation intervention in the UK major trauma pathway
AUTHORS	Kettlewell, Jade; Radford, Kate; Kendrick, Denise; Patel, Priya; Bridger, Kay; Kellezi, Blerina; dasNair, Roshan; Jones, Trevor; Timmons, Stephen

VERSION 1 – REVIEW

REVIEWER	Battle, Ceri Morrison Hospital, Welsh Institute of Biomedical and Emergency Medicine Research
REVIEW RETURNED	06-Feb-2022

GENERAL COMMENTS	Bmjopen-2021-060294 Factors affecting the implementation of a vocational rehabilitation intervention following major trauma: understanding the context for delivery. Thank you for inviting me to review this manuscript. The authors have completed a qualitative study investigating the factors affecting the implementation of a vocational rehabilitation intervention following major trauma. The purpose of the study is to gain an understanding of the context for delivery of the intervention, which will be investigated in a future trial. This is a very well written and conducted study and I have only a few comments / suggestions for the authors. INTRODUCTION: You provide an informative and succinct overview of the literature which sets the scene for the rest of the paper. METHODS: Page 6, line 1. Perhaps you should add a sentence stating the overall study design at the start of this section. Page 6, line 13. How did you identify your participants for the purposive sample? How were they invited to participate? Did you use incentives and did many refuse to participate? Page 6, lines 16-21. Why was consent needed for some of the data collection techniques, but not others? Page 7, lines 19-25. Excellent PPI involvement. Do you have diversity within your PPI group? RESULTS: Table 3: do you have any information to present regarding diversity, other than type of injury?
---

	Page 9, line 16: Structural characteristics: Please could you clarify for the reader that this refers to the employer's organisation, rather than the NHS organisation Page 14, line 12: Typo – you have written "...support RTW could become of barrier...". DISCUSSION: Page 17, line 3: You state you increased therapist knowledge about VR and providing in-depth training to explain the importance of early intervention in the trauma population". For your future trial, how will you achieve this on a wider scale, when you have more than 5 sites to consider? Page 18, line 6: What about sites where there is no psychologist within the service to deliver the intervention, to ensure both physical and mental health issues are considered during RTW? Page 18, line 15: Was there only one limitation of the study? What about the use of a purpose sample? Could that have introduced any bias? General question: What about patients who are self-employed? It is well known anecdotally (by clinicians) that patients who are self-employed often have very different needs and considerations re RTW? Did you consider this in your study?
--	--

REVIEWER	Dornonville de la Cour, Frederik Cervello, Research and Development
REVIEW RETURNED	11-Feb-2022

GENERAL COMMENTS	Thank you for inviting me to review this interesting manuscript. The study explores barriers and enablers for delivering vocational rehabilitation following traumatic injury based on interviews with a variety of stakeholders (at multiple centres) in the return-to-work process, except employers, and findings were used to inform adaptations to a VR intervention. I would like to commend the authors for conducting this comprehensive and important work. The manuscript is well written and concise, and I find that, in general, the procedures are described in a detailed and transparent way. Many perspectives are represented and discussed in this study, and the findings will be useful for providing an overview of the complex challenges in vocational rehabilitation across multiple health conditions to both researchers and clinicians/service providers. I have a few suggestions, which I hope the authors will consider. Page numbers refer to pages in the pdf (proof). 1) I find a minor inconsistency in the aims of the study and the focus of the discussion and conclusion sections. The discussion section and the conclusion are primarily concerned with adaptations to the ROWTATE intervention based on the findings of the study, while the aims were to investigate the context and the facilitators and barriers of delivering and implementing VR. Was it initially an aim of the study to inform and adapt the ROWTATE intervention? 2) Further, as a reader, I am interested to know what barriers the authors may expect to remain after adapting the intervention? Were all barriers targeted by adaptations to the program?
---

	3) P7 line 12-14: “purposive sampling”, please elaborate. What were the target number of participants, if any, and how did you assess the amount of data needed - or how did you arrive at the number of data collected? 4) How were the study participants approached and invited to the study? Were any participants reimbursed for their participation? Did anyone decline to participate, and if so, how may this impact generalisability? 5) Was the generation and structure of themes and subthemes guided by any theory or previous reading of the literature? 6) P15 line 18: The wording “good employer” could be revised for a more objective writing. 7a) Regarding generalisability, were any professions or other kind of stakeholders in vocational rehabilitation (except employers) not represented in the sample? Further, on P19 line 14-15, authors state that employers did not participate. How may this affect the findings and conclusions of the study? 7b) Also, some study participants described differences between major trauma centres and other services located further away. Do the authors expect the results of this study to generalise to other services outside the major trauma centres? Are the barriers and enablers likely to be the same or could there be any differences? 8) As disclosed, the study explores the perspectives of service providers and service users without investigating the perspectives of employers, and the authors argue that early contact with employers to increase awareness of the impact of injuries may facilitate RTW. Are the authors aware of previous research investigating the perspectives of employers in matters related to VR and RTW following injury? How may these perspectives, e.g., motivations and the agenda of employers, impact the recommendations for adaptations to the VR program? 9) Authors argue that current evaluations tend to focus on one health condition. I agree with the authors that many aspects of vocational rehabilitation are likely to be generalised across health conditions, however, some issues may be specific to the health condition in question. Did the authors identify any barriers specific to different types of injuries? E.g., injuries to the brain may involve cognitive deficits (e.g., anosognosia and personality changes), post-TBI fatigue, and other sequelae related to dysfunction of the central nervous system, which may cause barriers that are different to those of other types of injuries. 10) Reading through the manuscript, I noted a few typos, e.g. P5 lines 13-22 and P19 line 24, and I would recommend the authors to proof-read the manuscript. In addition, please provide a reference to the Injury Severity Score on P6 lines 9-10.
--	--

VERSION 1 – AUTHOR RESPONSE

Reviewer 1

Thank you for reviewing our manuscript and your positive comments.

INTRODUCTION:

1. You provide an informative and succinct overview of the literature which sets the scene for the rest of the paper.

Thank you for your comments, we are pleased that the introduction was clear and informative.

METHODS:

2. Page 6, line 1. Perhaps you should add a sentence stating the overall study design at the start of this section.

Thank you for your suggestion, the sentence has been changed to state:

'This multiple-methods qualitative study was part of a larger programme of work funded by the National Institute for Health Research (RP-PG-0617-20001).'

3. Page 6, line 13. How did you identify your participants for the purposive sample? How were they invited to participate? Did you use incentives and did many refuse to participate?

We have added the following for clarification:

'Potential participants were identified via known contacts of the authors at the different major trauma centres, who were then asked to share the invitation with relevant colleagues or trauma patients (i.e. snowball sampling). Service users were identified via trauma Patient and Public Involvement (PPI) groups at the university and via a database of past patient participants that had agreed to be contacted about future research.'

We did offer incentives and have added the following: 'Trauma participants and carers were offered a £20 gift voucher for their time, plus travel expenses where necessary.'

We did not collect information about who refused to participate, so cannot address this point.

4. Page 6, lines 16-21. Why was consent needed for some of the data collection techniques, but not others?

Thank you for your comment. We required consent for all data collection techniques, however written informed consent was required for interviews and focus groups, as these were audio recorded and transcribed. The research ethics committee did not require written informed consent for the co-design workshops, unless these were recorded, or for informal interviews. We have added the following to address your points: 'All participants were informed about the aim of the study and their right to withdraw. Written informed consent was obtained from all interviews and focus groups. For co-design workshops and walk-through care pathways, consent was assumed if participants did not opt out.'

5. Page 7, lines 19-25. Excellent PPI involvement. Do you have diversity within your PPI group?

Thank you for your positive comment. We have added a sentence to summarise the PPI group, which states, 'alongside 11 trauma survivors (diverse group including 7 males, 4 females; self-reported injury: spinal cord injury (n=2,) traumatic brain injury (n=8), multiple fractures (n=5), polytrauma (n=1); pre-injury occupation: professional (n=6), managerial (n=2), student (n=1), military (n=1).'

RESULTS:

6. Table 3: do you have any information to present regarding diversity, other than type of injury?

Thank you for your comment, we do have information about gender for all participants, and ethnicity and pre-injury occupation data for service users which we have added to the table.

7. Page 9, line 16: Structural characteristics: Please could you clarify for the reader that this refers to the employer's organisation, rather than the NHS organisation

Thank you for pointing out this error, the text has been amended to state: 'This theme refers to how the employing organisation works'

8. Page 14, line 12: Typo – you have written "...support RTW could become of barrier...".

We have amended to state: 'adjustments to support a RTW could become a barrier to VR intervention implementation'

DISCUSSION:

9. Page 17, line 3: You state you increased therapist knowledge about VR and providing in-depth training to explain the importance of early intervention in the trauma population". For your future trial, how will you achieve this on a wider scale, when you have more than 5 sites to consider?

We will provide a training and mentoring package to all occupational therapists and clinical psychologists delivering the intervention at all sites. We have clarified this by changing the text to read:

'We have sought to address cultural issues within the intervention design by developing an in-depth training and mentoring package to explain and provide evidence for the importance of a VR early intervention in the trauma population⁴³⁻⁴⁶ and increase therapist knowledge of VR. This training and mentoring package will be used for occupational therapists and clinical psychologists providing the intervention in the definitive trial.'

10. Page 18, line 6: What about sites where there is no psychologist within the service to deliver the intervention, to ensure both physical and mental health issues are considered during RTW?

We have clarified how we will deal with this in the definitive trial by adding the following text:

'For the definitive trial, where trauma services do not have access to psychological support, we will explore options for provision of psychological support from other NHS Trusts or from private practitioners.'

11. Page 18, line 15: Was there only one limitation of the study? What about the use of a purpose sample? Could that have introduced any bias?

Thank you for your comment. We have added further information about the rationale for our purpose sample, which we hope is sufficient:

'We chose this method to ensure we recruited key stakeholders within each major trauma centre that had a clear understanding of the rehabilitation pathways (based on a priori knowledge of clinical expertise in each site), and to ensure different injury types, pre-injury occupations (including self-employed), socioeconomic backgrounds, geographical locations, and different professionals (including employers) were represented in our sample.'

We ensured we recruited a range of stakeholders using a sampling framework to limit bias (e.g. to identify stakeholders with different injury types, expertise, professions etc.) and feel we had a sufficiently large sample size to enable us to present a broad perspective of stakeholders across the UK. Instead, we have added the following as a limitation:

'Although we aimed to engage with as many different stakeholders as possible, we do feel some professions were underrepresented in our sample, including nurses, prosthetists, orthotists and

trauma surgeons.'

12. General question: What about patients who are self-employed? It is well known anecdotally (by clinicians) that patients who are self-employed often have very different needs and considerations re RTW? Did you consider this in your study?

Thank you for mentioning this as it is an important point. We did recruit some self-employed participants as we wanted to ensure their views were explored. Additional demographic information has been included in Table 3 to show that we recruited 5 self-employed participants.

Reviewer 2

Thank you for reviewing our manuscript and your positive comments.

1) I find a minor inconsistency in the aims of the study and the focus of the discussion and conclusion sections. The discussion section and the conclusion are primarily concerned with adaptations to the ROWTATE intervention based on the findings of the study, while the aims were to investigate the context and the facilitators and barriers of delivering and implementing VR. Was it initially an aim of the study to inform and adapt the ROWTATE intervention?

Thank you for your comment. This is correct, the purpose of this stage of the programme grant was to explore implementation issues to inform the development of our intervention. We understand this is not clear in the aims, thus we have amended the aims to state:

'The findings of this study informed the development of the ROWTATE intervention and considerations for implementation ahead of a future trial.'

2) Further, as a reader, I am interested to know what barriers the authors may expect to remain after adapting the intervention? Were all barriers targeted by adaptations to the program?

Thank you for your comment, we have added the following to the discussion:

'We addressed as many issues as possible with the adaptations of the intervention (e.g. improving communication by including a central point of contact/case manager), however some implementation issues are not easily overcome (e.g. culture, organisational structure). Such barriers require long-term improvements, not only to change professional behaviours and cultural norms, but also to make small-step system changes to enhance coordination across the trauma pathway. Whilst we adapted our intervention to overcome known barriers, we anticipate new implementation issues arising when the intervention is delivered as part of a trial (i.e. real-world testing).'

3) P7 line 12-14: "purposive sampling", please elaborate. What were the target number of participants, if any, and how did you assess the amount of data needed - or how did you arrive at the number of data collected?

Thank you for your comments, we have added the following to the methods for clarification:

'We chose this method to ensure we recruited key stakeholders within each major trauma centre that had a clear understanding of the rehabilitation pathways (based on a priori knowledge of clinical expertise in each site), and to ensure different injury types, pre-injury occupations (including self-employed), socioeconomic backgrounds, geographical locations, and different professions (including employers) were represented in our sample.'

We planned to recruit up to 195 participants and stopped recruitment when we reached theoretical sufficiency at 117 participants. The methods have been edited to state:

'We aimed to recruit up to 195 participants (focus groups $n \leq 40$; interviews $n \leq 20$; walk-through care

pathways n≤60, up to 12 per site; workshops n≤75, up to 15 per site), however stopped recruitment when we reached theoretical sufficiency²². Due to the COVID-19 pandemic, we were unable to conduct as many walk-through care pathways interviews as proposed, thus conducted additional semi-structured interviews.'

4) How were the study participants approached and invited to the study? Were any participants reimbursed for their participation? Did anyone decline to participate, and if so, how may this impact generalisability?

We have addressed most of these comments in response to reviewer 1's comment number 3 above. Thank you for your comment about generalisability. The sampling strategy and characteristics of potential participants were outlined in a sampling framework (used to ensure diversity in our sample) which was developed through discussion within the large programme team, PPI, review of existing literature and experience from previous VR trials. Whilst we did not keep track of participants refusing to participate, attempts were made to recruit participants from each area of our sampling framework.

5) Was the generation and structure of themes and subthemes guided by any theory or previous reading of the literature?

The themes and subthemes were informed by the CFIR, which is mentioned in the text where we state: 'All data were thematically analysed using the framework approach and coded using NVivo (by JK & PP), informed by the Consolidated Framework for Implementation Research (CFIR)'. We have added the following for clarification:

'The main domains of CFIR were used for coding, and key themes were agreed by discussion with other authors...' We hope this provides sufficient information regarding the generation of themes.

6) P15 line 18: The wording "good employer" could be revised for a more objective writing. Thank you for your comment, this has now been reworded to state 'supportive employer'.

7) Regarding generalisability, were any professions or other kind of stakeholders in vocational rehabilitation (except employers) not represented in the sample?

Although we consulted with a wide range of stakeholders, not all healthcare professions were represented in our sample. We have acknowledged this as a limitation in our discussion by amending the text to read:

'However, we were unable to recruit any employers to participate in the interviews or focus groups, due to the COVID-19 pandemic and difficulty contacting employing organisations. This meant the views of these stakeholders were not explored, and further work is required to understand their important perspectives on RTW post-trauma. Although we aimed to engage with as many different stakeholders as possible, we do feel some professions were underrepresented in our sample, including nurses, prosthetists, orthotists and trauma surgeons.'

8) Further, on P19 line 14-15, authors state that employers did not participate. How may this affect the findings and conclusions of the study?

This has been addressed by our response to the reviewer's last comment.

9) Also, some study participants described differences between major trauma centres and other services located further away. Do the authors expect the results of this study to generalise to other services outside the major trauma centres? Are the barriers and enablers likely to be the same or could there be any differences?

Thank you for your comment. The following has been added to the end of the conclusion, 'Findings could illuminate similar barriers in other complex healthcare pathways outside of major trauma (e.g., cultural norms, poor communication), and may inform intervention implementation within other

conditions.'

9) As disclosed, the study explores the perspectives of service providers and service users without investigating the perspectives of employers, and the authors argue that early contact with employers to increase awareness of the impact of injuries may facilitate RTW. Are the authors aware of previous research investigating the perspectives of employers in matters related to VR and RTW following injury? How may these perspectives, e.g., motivations and the agenda of employers, impact the recommendations for adaptations to the VR program?

Thank you for your comments. We are aware of previous research exploring the perspectives of employers supporting return-to-work in the acquired brain injury population. Although we were unable to explore the views of employers across a broader trauma population, evidence suggests that employers and patients are important stakeholders in a successful return-to-work and both should be actively involved in the process.

We have added the following to the discussion to acknowledge this evidence:

'We drew on existing evidence to develop the employer engagement component of our intervention^{51, 52}. Previous research in acquired brain injury and spinal cord injury indicates that patients who understand the impact of their injuries are more able to discuss their limitations with employers^{17, 53}. Understanding their employee's injury and limitations is essential in facilitating a successful RTW^{17, 53, 54}. In general, employers feel they lack the necessary experience and knowledge to support a RTW for someone post-injury^{17, 53}. Further evidence suggests that employers require functional advice (e.g. planning phased RTW, reasonable adjustments) as well as psychological support to understand and address the needs of their employee^{53, 54}. Providing a central point of contact to liaise with, educate and support the employer and patient, negotiating RTW plans, and providing advice and emotional support (via OT and clinical psychologist), are key components our intervention.'

10) Authors argue that current evaluations tend to focus on one health condition. I agree with the authors that many aspects of vocational rehabilitation are likely to be generalised across health conditions, however, some issues may be specific to the health condition in question. Did the authors identify any barriers specific to different types of injuries? E.g., injuries to the brain may involve cognitive deficits (e.g., anosognosia and personality changes), post-TBI fatigue, and other sequelae related to dysfunction of the central nervous system, which may cause barriers that are different to those of other types of injuries.

Thank you for your comment. We did identify issues specific to some injury types, however this has been described in an earlier publication from this study (Kettlewell, J. et al. A study of mapping usual care and unmet need for vocational rehabilitation and psychological support following major trauma in five health districts in the UK. *Clinical rehabilitation* 35.5 (2021): 750-764), so has not been repeated in this paper. Our findings showed that while there are many examples of rehabilitation identified especially for specific injury groups (e.g. traumatic brain and spinal injuries), our research shows there are many gaps in service provision, which were more pronounced for other injury groups (e.g. musculoskeletal injuries and amputations) and for patients located further from major trauma centres. The gaps and/or inconsistencies in care were especially problematic in relation to vocational rehabilitation and psychological services across the major trauma networks.

11) Reading through the manuscript, I noted a few typos, e.g. P5 lines 13-22 and P19 line 24, and I would recommend the authors to proof-read the manuscript. In addition, please provide a reference to the Injury Severity Score on P6 lines 9-10.

Thank you for noting the typos, the manuscript has been read thoroughly and all errors corrected. A reference has been added for the Injury Severity Score.

VERSION 2 – REVIEW

REVIEWER	Battle, Ceri Morrison Hospital, Welsh Institute of Biomedical and Emergency Medicine Research
REVIEW RETURNED	28-Feb-2022

GENERAL COMMENTS	Thank you for considering and responding to my previous comments. I have no further feedback, other than to congratulate the authors on their work.
---

REVIEWER	Dornonville de la Cour, Frederik Cervello, Research and Development
REVIEW RETURNED	02-Mar-2022

GENERAL COMMENTS	Thank you for the opportunity to review your resubmission entitled: "A qualitative study exploring factors affecting the implementation of a vocational rehabilitation intervention in the UK major trauma pathways". The authors have addressed all of my previous comments adequately, and I was satisfied with the changes you have made to the revised manuscript, which added clarity. I have no further comments after reviewing the manuscript, except for a few minor amendments (page numbers refer to page numbers in the manuscript): P18 line 17: A period is lacking after "limitations with employers17, 53" and the following sentence should be corrected from "Understand..." to "Understanding..." P18 line 24: The word "of" is lacking in "are key components our intervention." P19 line 21: Please correct "Our research team is comprised practitioners..."
---